# Multiple Cylinder Extraction from Organized Point Clouds

**DOI:** 10.3390/s21227630

**Published:** 2021-11-17

**Authors:** Saed Moradi, Denis Laurendeau, Clement Gosselin

**Affiliations:** 1Department of Electrical and Computer Engineering, Faculty of Science and Engineering, Laval University, Quebec, QC G1V0A6, Canada; saed.moradi@gmail.com; 2Computer Vision and Systems Laboratory (CVSL), Laval University, Quebec, QC G1V0A6, Canada; 3Robotics Laboratory, Department of Mechanical Engineering, Faculty of Science and Engineering, Laval University, Quebec, QC G1V0A6, Canada; clement.gosselin@gmc.ulaval.ca

**Keywords:** organized point clouds, depth map, surface normal estimation, cylinder detection, cylinder extraction

## Abstract

Most man-made objects are composed of a few basic geometric primitives (GPs) such as spheres, cylinders, planes, ellipsoids, or cones. Thus, the object recognition problem can be considered as one of geometric primitives extraction. Among the different geometric primitives, cylinders are the most frequently used GPs in real-world scenes. Therefore, cylinder detection and extraction are of great importance in 3D computer vision. Despite the rapid progress of cylinder detection algorithms, there are still two open problems in this area. First, a robust strategy is needed for the initial sample selection component of the cylinder extraction module. Second, detecting multiple cylinders simultaneously has not yet been investigated in depth. In this paper, a robust solution is provided to address these problems. The proposed solution is divided into three sub-modules. The first sub-module is a fast and accurate normal vector estimation algorithm from raw depth images. With the estimation method, a closed-form solution is provided for computing the normal vector at each point. The second sub-module benefits from the maximally stable extremal regions (MSER) feature detector to simultaneously detect cylinders present in the scene. Finally, the detected cylinders are extracted using the proposed cylinder extraction algorithm. Quantitative and qualitative results show that the proposed algorithm outperforms the baseline algorithms in each of the following areas: normal estimation, cylinder detection, and cylinder extraction.

## 1. Introduction

The rapid development of three-dimensional (3D) scanning devices has provided a unique opportunity for robotic applications to effectively interact with the real world. Object grasping, as a common robotic task, has attracted the attention of researchers during the past decade. Most man-made objects are composed of a few geometric primitives (GPs) such as spheres, cylinders, planes, ellipsoids, or cones. Thus, object recognition can be reduced to a problem of geometric primitives extraction. Among the different geometric primitives, cylinders are the most frequently used GPs in real-word scenes [1]. Also, cylinder detection and extraction are used in several industrial applications like pipeline plant modeling [2], reverse engineering [3], automatic forest inventory [4], and 3D facility modeling [5]. This is why cylinder detection and extraction is of great importance in 3D computer vision.

There are several approaches dedicated to cylinder extraction in the literature. Hough-based methods [6,7,8], RANdom SAmpling Consensus (RANSAC)-based methods [1,9,10,11], Robust PCA [12], and quadric fitting [13] are the most popular ones for point cloud data. In [12] the authors proposed a cylinder fitting method in laser scanning point cloud data based on robust principal components analysis. After decomposition, the cylinder orientation is estimated based on the principal component corresponding to the largest eigen-value (PC1), and PC2 as well as PC3 are used for the identification of the radius and center of the cylinder. Authors in [14] used a soft voting scheme based on curvature information to exclude outliers from cylindrical parts. In order to further remove outliers, they trained a deep-learning based classifier to filter them out. In [15], the authors projected the point cloud onto a set of directions over the unit hemisphere and tried to detect circular projections. Then the cylindrical surfaces are extracted by fitting a cylinder to each connected component. The authors in [16] used several cylinder cutting planes to obtain different ellipses. Then, RANSAC is used for both ellipse and cylinder fitting. Automated 3D pipelines recognition is investigated in [17]. The principal curvature is used as a cylinder detection algorithm and the parameters of the cylinder are extracted using RANSAC algorithm. While there is a rich literature on single cylinder extraction, multiple cylinder detection has not been investigated in depth.

In this paper, a fast and robust method is presented for multiple cylinder extraction from point cloud data. The rest of this paper is organized as follows: The research problem is defined in Section 2. In Section 3, novel solutions are proposed for the problem of multiple cylinder extraction in point cloud data. The effectiveness of the proposed solution is investigated in Section 4. Finally, the paper is concluded in Section 5.

## 2. Problem Definition

The problem of a cylinder extraction in point cloud data can be divided into three successive steps. At the first step, the input point cloud (3D points in Cartesian coordinates) may be represented in another feature space to deal with useful information. The orientation of surface normal vectors is one of the most common features used in 3D data processing [18,19,20,21]. There have been significant efforts dedicated to normal estimation from point cloud data in the literature. However, there is still a lack of fast and robust methods for normal estimation. This problem is investigated in depth in Section 2.1. Thereafter, the problem regarding cylinder detection and extraction is described in Section 2.2.

### 2.1. Normal Estimation from Point Clouds

Normal vectors estimation [22,23,24] is the cornerstone of many 3D computer vision tasks such as segmentation [25], registration [26], surface construction [27], object recognition [28], and others. The most common approach to estimate the surface normal vector at a point is to fit a plane to a local neighborhood of the query point and determine the vector normal to the tangent plane (See Appendix A). Numerous efforts have been made over the last decade to improve the accuracy of the surface normal estimation for unorganized point clouds. In [29], robust statistics are used to fit the optimum tangent plane for points located on high curvature surfaces. Boulch et al. [30] used Randomized Hough Transform (RHT) with statistical exploration bounds to preserve sharp features robustly. They also used a fixed-size accumulator to decrease the execution time of the estimation process. Liu et al. [31] took advantage of the results of tensor voting to decrease the estimation error. Since their algorithm has huge computational complexity, they used a GPU-based implementation in order to meet the requirements of real-time processing. In [32], the Deterministic MM-estimator (DetMM) is used to exclude outlier points for robust normal estimation. In addition to classical data processing techniques, deep learning-based methods have recently attracted the attention of the research community for surface normal vector estimation [22,33].

While many research studies dedicated to normal estimation for unorganized point clouds are reported in the literature, normal estimation directly from a depth map (organized point cloud) has received less attention. Computing normal vectors from depth images has the following advantages:

The points in the local neighborhood of the point for which the normal is computed are known while for unorganized point clouds, an extra processing step is needed to determine the points belonging to this neighborhood.Most operations on organized point clouds can be performed using 2D operators and they are generally faster than 3D operations.The normal vectors can be computed during the scanning process.

Despite the advantages of normal estimation from depth maps, there are still some challenges. First, the input depth image is contaminated by measurement noise. Also, sharp depth discontinuities in the image can reduce the robustness of the estimated normal vectors. Fortunately, due to recent advances in scanning technology, high-quality scanners are available even at the consumer-grade level. Thus, by following a proper strategy, normal vectors can be computed directly from a raw depth map in a fast and robust manner.

Some research in the literature focuses on the estimation of normal vectors directly from input depth images. Tang et al. [34] proposed a closed-form solution for normal vectors at each point. However, erroneous formulation in tangent vectors and approximation of first-order derivatives led to a poor result in terms of accuracy. Holzer et al. [35] used an adaptive neighboring size for each point to increase the accuracy of normal estimation in sharp edges. They also used integral images for the sake of computational efficiency. The first problem related to their method is that no analytical method for the determination of the design parameters is reported and the parameters are simply computed empirically. Their method has large errors for small objects with high surface curvature. One of the most accurate normal estimation methods from a depth map was presented by Nakagawa et al. [36]. While the tangent vector construction is performed accurately, the use of single-pixel padded approximation of first-order partial derivatives and propagation of that error after a cross product of tangent vectors led to a poor and noisy result. In Section 3, we propose a fast and robust method for the estimation of normal vectors from raw depth maps that addresses all of the above issues.

### 2.2. Cylinder Detection and Extraction

The previously presented research works for cylinder detection and extraction suffer from two main drawbacks:They are only able to detect (or extract) one cylinder at a time. The points belonging to each detected cylinder must be removed from the point cloud before starting the detection process of the next cylinder.The success or failure of the detection algorithm strongly depends on the initial point (seed) selection. Without a proper strategy for robust initial point selection, the main cylinder detection module often fails.

Depth measurement errors are present in images captured by 3D sensors including the Microsoft Kinect [37]. These errors, especially outliers, are more often present near object boundaries and affect the resulting point clouds. When a local neighborhood is constructed around a query point in order to find a geometric primitive, it should not include these outliers to avoid erroneous extraction. For instance, Figure 1 shows examples of good and poor areas for initial sample selection by a cylinder extraction algorithm. The green points are good candidates for initial sample selection, while choosing the seed point among red ones may result in failure of the cylinder extraction process. Therefore, an effective cylinder detection algorithm should be able to robustly detect good initial points and reject poor ones. As shown in Figure 1, the proper candidate points for initial sample selection are located far from object boundaries.

In addition to the problem relevant to the initial sample selection, multiple cylinder detection has also not been investigated in depth in the literature. Generally, a recursive process is used to detect (or extract) all of the cylinders present in the scene. To this end, after extracting the first cylinder, the points belonging to this cylinder are removed from the scene and the detection process is started over until there are no cylinders remaining in the scene. The main drawbacks of this approach are:The whole process must be repeated for each cylinder. The overall execution time is increased drastically when many cylinders should be extracted in the scene.Since the whole point cloud is modified after each detection step, the previous computations may not be reusable.The initial sample selection criterion should be met at each detection step.

In the next section, an approach circumventing these problems is proposed for multiple cylinder extraction.

## 3. Proposed Solution

In order to address the aforementioned problems, a fast and robust algorithm is presented for the detection and extraction of multiple cylinders in organized point clouds. The overall procedure of the proposed method is illustrated in Figure 2.

As shown in Figure 2, in the first step, 3D surface normal vectors are estimated using a novel fast and accurate algorithm from raw input depth maps. The normal vectors are then represented in spherical coordinates to produce more distinguishable surface features. Since the normal vectors have a unit length, the third component in spherical coordinates ($I_{r}$) does not contain useful information and can be discarded. Thus, the normal vectors for all points can be represented using a pair of images ($I_{\phi }$ and $I_{\theta }$) containing the orientation angles. Both images contain useful features to distinguish cylindrical surfaces from the rest of the scene. In the next step, the Maximally Stable Extremal Regions (MSER) feature detector is used to detect cylinders in the scene. In the final step, a fast cylinder extraction approach is proposed to estimate the parameters (axis direction and radius) of each cylinder. In the following, each of these three proposed sub-modules is explained in detail.

### 3.1. Fast Surface Normal Estimation

A depth image d=g(r,c) can be converted to an organized point cloud PC(x,y,z) using camera calibration information (Figure 3). The following equations are used for this conversion:(1)$$\begin{array}{c} z=d,\\ x=\left(r-o_{x}\right)⊙\left(\frac{d}{f_{x}}\right),\\ y=\left(c-o_{y}\right)⊙\left(\frac{d}{f_{y}}\right), \end{array}$$
where $o_{x}$ and $o_{y}$ are the coordinates of the principal points (the optical center), $f_{x}$ and $f_{y}$ denote the focal lengths, and ⊙ stands for element-wise multiplication. Therefore, every single pixel in a depth map corresponds to a position in 3D world. The most trivial way to estimate the surface normal vector is to compute the cross product of two perpendicular tangent vectors. Considering a smooth surface, the tangent vectors can be constructed from a depth map as shown in Figure 4. Thus, the surface normal vector is expressed as:(2)$$n=\left[\begin{array}{c} n_{x}\\ n_{y}\\ n_{z} \end{array}\right]=s_{12}\times s_{13}$$
where, $s_{12}$ and $s_{13}$ are two tangent vectors and determined as:(3)$$s_{12}=\left[\begin{array}{c} x_{2}-x_{1}\\ y_{2}-y_{1}\\ z_{2}-z_{1} \end{array}\right]=\left[\begin{array}{c} u_{2}d_{2}-u_{1}d_{1}\\ v_{2}d_{2}-v_{1}d_{1}\\ d_{2}-d_{1} \end{array}\right]\;,\;s_{13}=\left[\begin{array}{c} x_{3}-x_{1}\\ y_{3}-y_{1}\\ z_{3}-z_{1} \end{array}\right]=\left[\begin{array}{c} u_{3}d_{3}-u_{1}d_{1}\\ v_{3}d_{3}-v_{1}d_{1}\\ d_{3}-d_{1} \end{array}\right]$$
where, $d_{i}=g\left(r_{i},c_{i}\right)$, $u_{i}=\frac{r_{i}-o_{x}}{f_{x}}$, $v_{i}=\frac{c_{i}-o_{y}}{f_{y}}$. According to Figure 4, the following equations hold:(4)$$v_{2}=v_{1}\;,\;u_{1}=u_{3}\;,\;v_{3}=v_{1}+\frac{\alpha }{f_{y}}\;,\;u_{2}=u_{1}+\frac{\alpha }{f_{x}}$$

In Equation (Equation 4), α denotes the pixel distance between two points (Figure 4).

Using Equation (Equation 2), each component of the normal vectors can be separately calculated as follows:(5)$$n_{x}=\left(v_{2}d_{2}-v_{1}d_{1}\right)\left(d_{3}-d_{1}\right)-\left(v_{3}d_{3}-v_{1}d_{1}\right)\left(d_{2}-d_{1}\right)$$
(6)$$n_{y}=\left(u_{3}d_{3}-u_{1}d_{1}\right)\left(d_{2}-d_{1}\right)-\left(u_{2}d_{2}-u_{1}d_{1}\right)\left(d_{3}-d_{1}\right)$$
(7)$$n_{z}=\left(u_{2}d_{2}-u_{1}d_{1}\right)\left(v_{3}d_{3}-v_{1}d_{1}\right)-\left(v_{2}d_{2}-v_{1}d_{1}\right)\left(u_{3}d_{3}-u_{1}d_{1}\right)$$

Equation (Equation 5), Equation (Equation 6) and Equation (Equation 7) can be simplified using auxiliary equations in Equation (Equation 4). The final expressions are:(8)$$n_{x}=-\frac{\alpha }{f_{y}}d_{3}\left(d_{2}-d_{1}\right)$$
(9)$$n_{y}=-\frac{\alpha }{f_{x}}d_{2}\left(d_{3}-d_{1}\right)$$
(10)$$n_{z}=\frac{\alpha }{f_{x}}v_{1}d_{2}\left(d_{3}-d_{1}\right)+\frac{\alpha }{f_{y}}u_{1}d_{3}\left(d_{2}-d_{1}\right)+\frac{\alpha^{2}}{f_{x}f_{y}}d_{2}d_{3}$$

In case of noisy input, the averaging process on multi-scale (using different distance values α) results will suppress the noise effect. Therefore:(11)$$n_{x}=\frac{1}{K}\sum_{i=1}^{K}n_{x_{i}}\;n_{x_{i}}=-\frac{\alpha_{i}}{f_{y}}d_{3}\left(d_{2}-d_{1}\right)\;i=1,2,\cdots ,K$$
(12)$$n_{y}=\frac{1}{K}\sum_{i=1}^{K}n_{y_{i}}\;n_{y_{i}}=-\frac{\alpha_{i}}{f_{x}}d_{2}\left(d_{3}-d_{1}\right)\;i=1,2,\cdots ,K$$
(13)$$n_{z}=\frac{1}{K}\sum_{i=1}^{K}n_{z_{i}}\;n_{z_{i}}=\frac{\alpha_{i}}{f_{x}}v_{1}d_{2}\left(d_{3}-d_{1}\right)+\frac{\alpha_{i}}{f_{y}}u_{1}d_{3}\left(d_{2}-d_{1}\right)+\frac{\alpha_{i}^{2}}{f_{x}f_{y}}d_{2}d_{3}\;i=1,2,\cdots ,K$$

Finally, the orientation angles $I_{\phi }$ and $I_{\theta }$ can be calculated as:(14)$$I_{\phi }=\tan^{-1}\left(\frac{n_{y}}{n_{x}}\right)$$
(15)$$I_{\theta }=\tan^{-1}\left(\frac{\sqrt{n_{x}^{2}+n_{y}^{2}}}{n_{z}}\right)$$

### 3.2. Cylinder Detection Module

The orientation of the normal vectors is a useful extrinsic surface feature to distinguish among different types of geometric primitives. Typically, the dynamic range of the different orientation angles is split into non-overlapping bins. Then, the histogram of normals with these angles is used as the feature vector to train a classifier [18,21]. In this work, instead of the histogram of the orientation of the normal vectors, the different angles are considered as different *images*. To this end, the organized normal vectors $\left(n_{x},n_{y},n_{z}\right)$ in Cartesian coordinates are converted to spherical coordinates $(I_{r},I_{\phi },I_{\theta })$. Since the surface normals have a unit length, the first component of the normal vector in spherical coordinates does not hold useful information concerning the surface geometry. Therefore, the two remaining components $(I_{\phi },I_{\theta })$ are used for further processing. Figure 5 shows both $I_{\phi }$ and $I_{\theta }$ images of a sample depth image. As depicted in Figure 5c,d, both $I_{\phi }$ and $I_{\theta }$ images contain relevant information that can be used to distinguish between cylindrical and non-cylindrical areas. Cylindrical surfaces appear as a maximally stable elliptical region in the $I_{\theta }$ image, while there are sharp edges along the symmetry line of each cylinder in the $I_{\phi }$ image. Considering these two observations, a new cylinder detection approach is presented in the following.

The elliptical region in the $I_{\theta }$ image can be easily detected using the maximally stable extremal regions (MSER) feature detector due to following reasons:The good sample points (see Figure 1 for the definition of good sample points) of the cylindrical surfaces have a small deviation of θ (in the $I_{\theta }$ image). Thus, regions belonging to a cylindrical surface remain stable over a certain range of threshold values.Since there are small angular differences between the Z-axis and the normal vectors of good sample points, these points lie on local maxima pixels in the $I_{\theta }$ image.

Therefore, these regions are both stable and extremal which can efficiently be found using MSER feature detection (In this work, a MATLAB built-in function is used to detect MSER features and obtain MSER regions. The function returns a pixel list and orientation of each region). Figure 6 shows the detected MSERs in the $I_{\theta }$ image. In case of a false response detection, the $I_{\phi }$ image can be used to refine the results.

### 3.3. Cylinder Extraction Sub-Module

The most straightforward method of extracting a cylinder from a set of inlier points in a point cloud is proposed by Tran et al. [1]. Their approach consists of two steps. In the first step, the orientation axis of the cylinder is found. Then, all of the inlier points are projected onto a 2D plane normal to the orientation axis. Thus, the cylinder extraction problem is relaxed into a 2D circle fitting. In Tran’s method, the vector being the most orthogonal to all of the normal vectors is considered as the orientation axis of the cylinder. Identifying the orientation axis using this method requires the construction of a scattering matrix and eigenvalue decomposition which increases the execution time of the algorithm. In our method, we rather use the result of the detection step to identify the orientation axis of each cylinder. The process is described in Figure 7. As shown in the figure, the rotation angle of the MSER ellipse resulting from the detection step is used to rotate the MSER patch into a vertical patch. Then, a morphological dilation operator is used to fill holes in the MSER patch. After determining the bounding box for the region, two points on the line that passes across the bounding box are chosen as the start and the end points of the orientation axis. The reminder of the extraction process consists in determining the parameters of the projected circle in the 2D space of the plane normal to orientation axis. In this work, after a distance-based outlier removal, the Kåsa method is used for circle fitting [38].

## 4. Results

In order to evaluate the cylinder extraction performance of the proposed method, some experiments were carried out on real data captured by a Microsoft Kinect Azure RGB-D camera. The comparisons and evaluations are divided into two parts. The first part (Section 4.1) is dedicated to the normal vector estimation results. Detection and extraction of the cylinders in the scene are presented in the second part (Section 4.2).

### 4.1. Normal Estimation

In this section, the surface normal estimation results are presented and discussed. Figure 8 and Figure 9 show qualitative comparisons of the different methods. Since there is no ground truth data, the normal estimation results using the local plane fitting approach is used as the ground truth. As shown in the figures, Tang’s method achieves the worst performance. The $I_{\theta }$ image (Figure 8) is not constructed correctly using this method, and the $I_{\phi }$ image is very noisy (Figure 9) compared to the other algorithms. Nakagawa’s method achieves the second best performance among all of the algorithms. Both $I_{\theta }$ and $I_{\phi }$ components of the normal vectors have an acceptable quality. In summary, the normal estimation method proposed in this paper outperforms other baseline algorithms. As shown in the figures, both $I_{\theta }$ and $I_{\phi }$ components have the most similarity to the ground truth. The quantitative results of the Mean Squared Errors (MSE) and Structural SIMilarity (SSIM) index prove the effectiveness of the proposed normal estimation method (Table 1, Table 2, Table 3 and Table 4).

In order to provide a quantitative comparison in term of computational efficiency, all of the algorithms were implemented in MATLAB. The full specifications of the simulation environment are reported in Table 5. Table 6 shows the execution time for each algorithm. As reported in the table, the proposed algorithm is the second fastest algorithm after Tang’s method. Based on these results, the normal estimation process can be performed at 66 frames per second on a CPU-based implementation on a not so recent computer (See Table 5).

### 4.2. Cylinder Detection and Extraction

In order to compare the results of cylinder detection by the different algorithms, the Radius-based Surface Descriptors (RSD) [39] and mean and Gaussian curvature-based method [1] are used as baseline algorithms. The cylinder detection results are depicted in Figure 10. As shown in the figure, both baseline algorithms fail in detecting the cylinders in the scene, while the proposed method detected them correctly without missing any.

Since there are no ground truth for the orientation axis of the cylinders, we used a simple approach to compare our method to Tran’s method [1]. The object under test consists of two co-centered cylinder (Figure 11a). After points projection, the two circles corresponding to these cylinders must be clearly distinguishable. However, as depicted in Figure 11b Tran’s method fails in this way, while our method has a better performance (Figure 11c). Both methods performed well in radius estimation. With the exception of two objects where the measurements were not accurate enough due to their material (Object 1 is made of highly reflective plastic) and dimensions (Points belonging to long objects which are comparable to vertical field of view are not measured accurately), the extraction process performed satisfactorily in other cases (Table 7).

The complete cylinder extraction process from a raw depth image is demonstrated in Figure 12. The first and second columns of the figure show RGB and depth images of the input scene. As shown in the third column of the figure, all of the cylindrical surfaces are detected correctly.

## 5. Conclusions

In this paper, a new method is proposed to extract multiple cylinders from organized point clouds. Unlike the majority of the algorithms which focused on single cylinder extraction from a point cloud, all of the cylinders can be extracted from the scene using our method. Prior to the extraction step, a detection module is required to determine all of the points belonging to the different cylindrical surfaces. Since the initial sample selection plays a critical role in extracting the cylinder parameters, a straightforward approach is presented to this end, too. The overall extraction procedure consists of three sub-modules. The first module attempts to accurately estimate the normal vectors directly from the 2D depth map. Compared to the existing normal estimation methods, the proposed method can estimate the vectors in a fast and accurate manner. Considering each component of the normal vectors as an image, MSE and SSIM metrics are used to evaluate the accuracy of the proposed normal vector estimation method. Also, for a single-scale construction, the estimation method can run at 66 frame/sec rate.

After representation of the normal vectors in the spherical coordinates, a cylinder detection algorithm is proposed based on MSER feature detectors. This approach not only detects the existing cylinders in the scene simultaneously, but it also gives a pool of proper candidates (good sample points) for initial sample selection. The orientation of the query cylinder can then be easily identified using two candidate points from the pool. After identification of the orientation axis, all good sample points are projected onto a plane along the orientation axis. Using this approach, the complex cylinder fitting problem can be solved using 2D circle fitting approaches. The extraction results on real depth maps collected during this project demonstrate the effectiveness of the proposed algorithm for real-world applications.

## Figures and Tables

**Figure 1 sensors-21-07630-f001:**
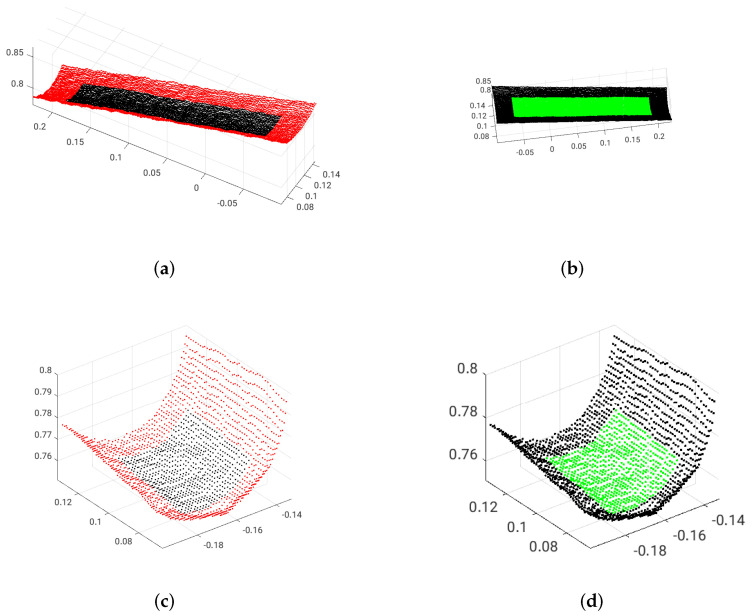
Examples of poor and good points for initial sample selection by a cylinder extraction algorithm. (**a**) poor initial surface points for object 1, (**b**) good initial surface points for object 1, (**c**) poor initial surface points for object 2, (**d**) good initial surface points for object 2. Red points are not good candidates for primitives extraction while green points lead to a successful extraction.

**Figure 2 sensors-21-07630-f002:**
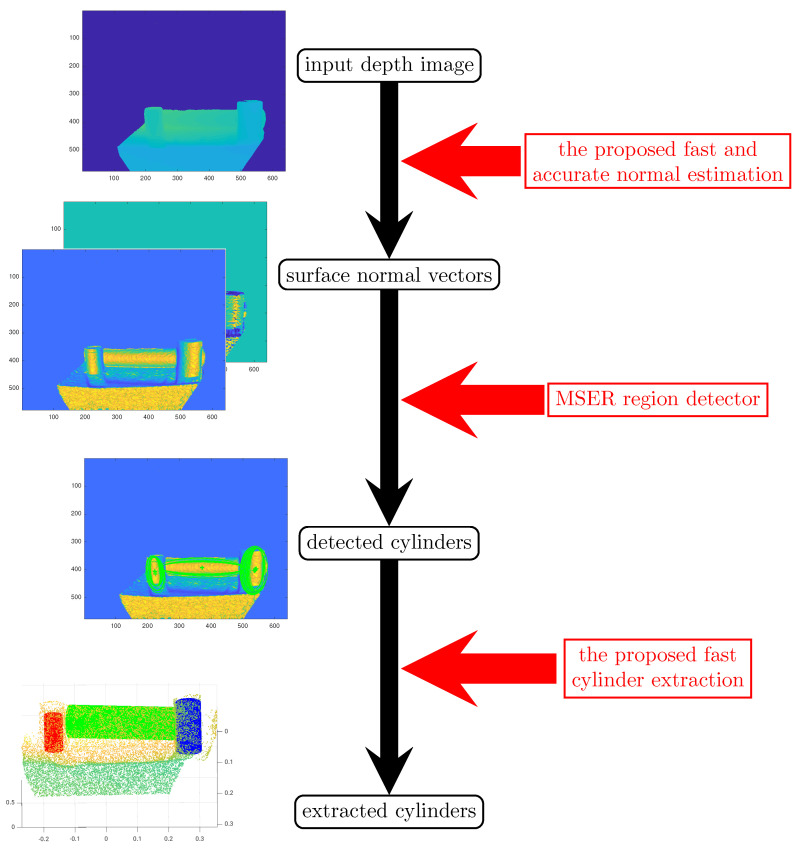
The overall procedure of the proposed method. The black text color indicates the data-type at each step. The required processing at each step is indicated in red.

**Figure 3 sensors-21-07630-f003:**
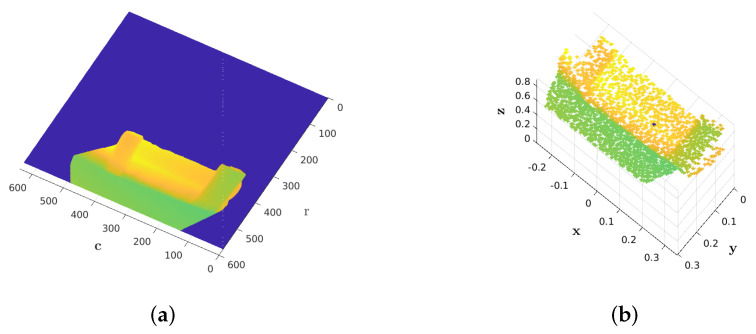
Correspondence between a depth map (**a**) and an organized point cloud (**b**).

**Figure 4 sensors-21-07630-f004:**
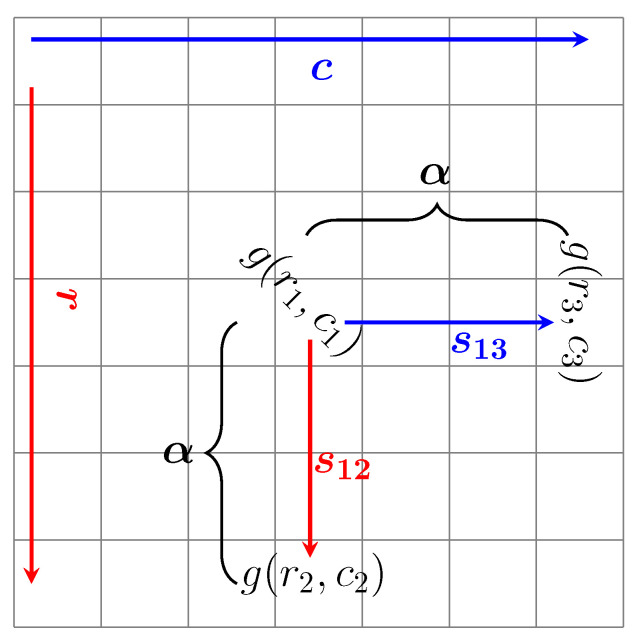
Surface tangent vectors construction from the depth map.

**Figure 5 sensors-21-07630-f005:**
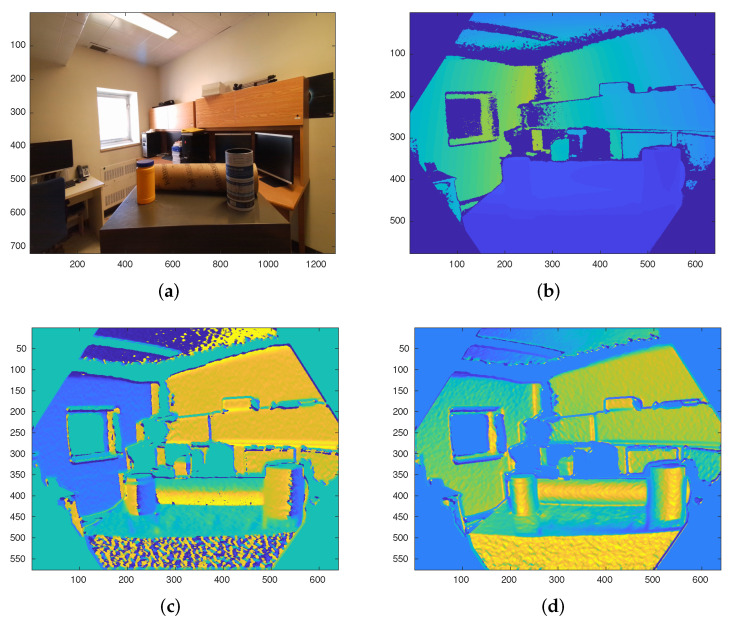
Different images from the same scene. (**a**) RGB, (**b**) Depth, (**c**) $I_{\phi }$, and (**d**) $I_{\theta }$ images.

**Figure 6 sensors-21-07630-f006:**
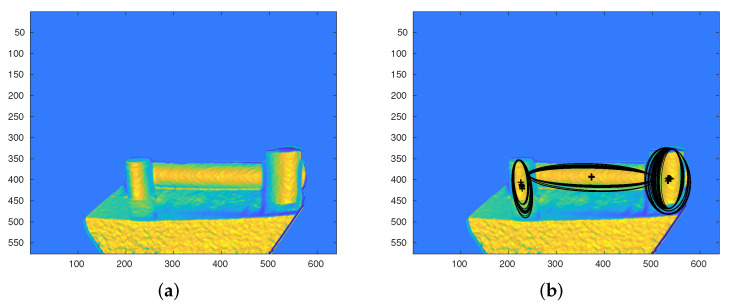
(**a**) $I_{\theta }$ image after applying a range filter, (**b**) detected MSERs.

**Figure 7 sensors-21-07630-f007:**
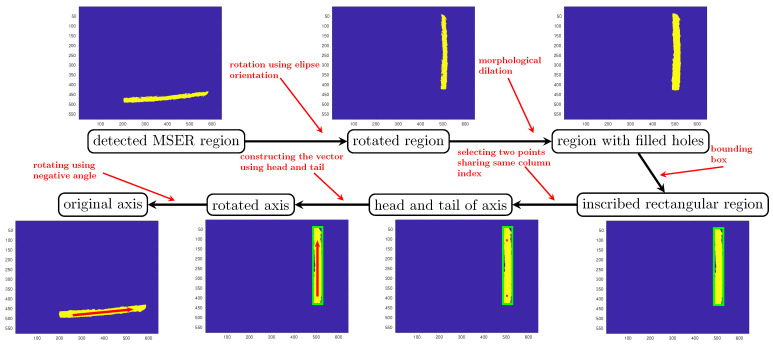
Estimating the orientation of the axis of a cylinder.

**Figure 8 sensors-21-07630-f008:**
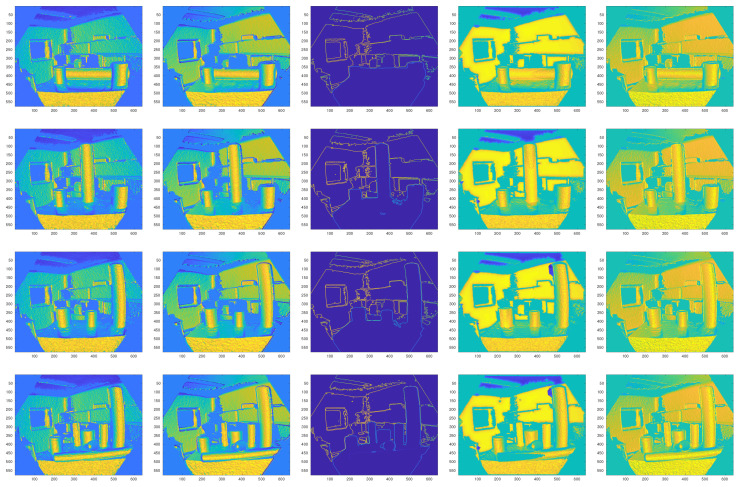
$I_{\theta }$ images of the estimation results of different algorithms. From left: the ground truth, our method, Tang’s method [34], Holzer’s method [35], Nakagawa’s method [36].

**Figure 9 sensors-21-07630-f009:**
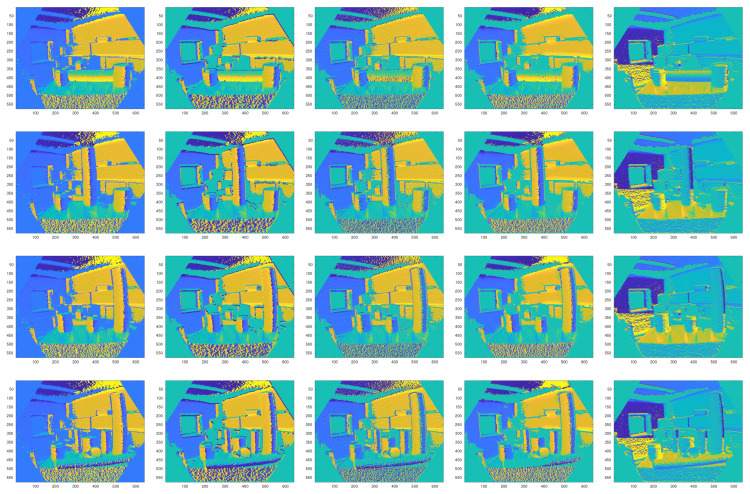
$I_{\phi }$ images of the estimation results of different algorithms. From left: the ground truth, our method, Tang’s method [34], Holzer’s method [35], Nakagawa’s method [36].

**Figure 10 sensors-21-07630-f010:**
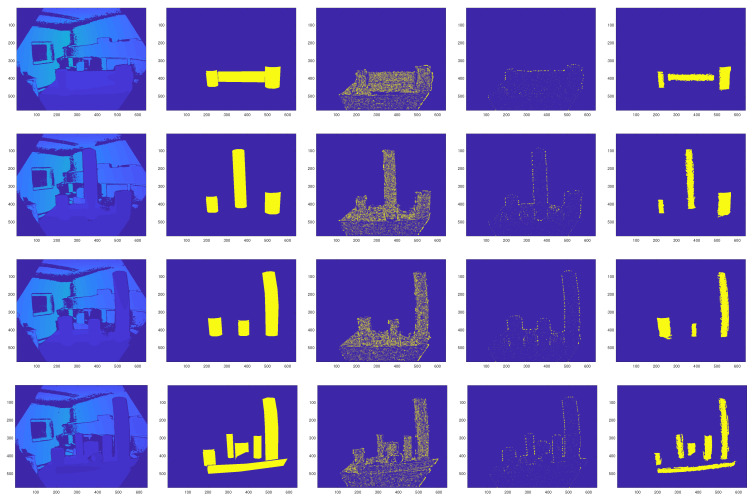
Cylinder segmentation results using different algorithms. From left: input depth image, ground truth, RSD-based method, principal curvature-based method, our method.

**Figure 11 sensors-21-07630-f011:**
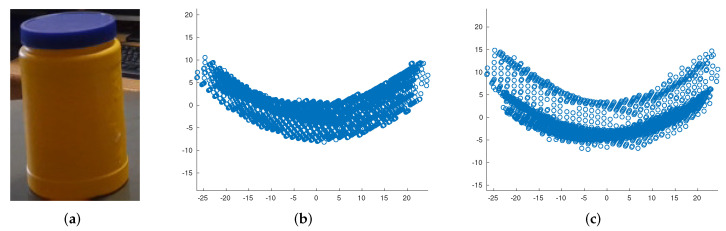
Orientation axis identification of the cylinders. (**a**) The cylindrical object, (**b**) 2D projected points resulting from Tran’s method, (**c**) 2D projected points resulting from our method.

**Figure 12 sensors-21-07630-f012:**
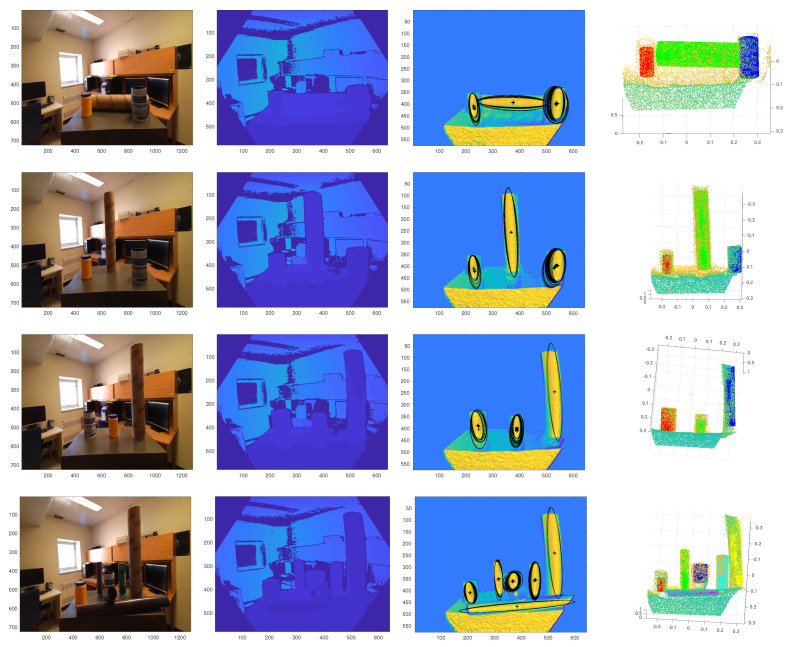
The first column: RGB image, the second column: depth image, the third column: detected MSERs, the fourth column: final cylinder extraction results. For our method, all cylinders are detected and extracted simultaneously.

**Table 1 sensors-21-07630-t001:** Mean Squared error (MSE) of $I_{\theta }$ images.

	Tang’s Method	Holzer’s Method	Nakagawa’s Method	Ours
1st scene	1.1384	1.6177	1.4101	**0.1070**
2nd scene	1.1324	1.4699	1.4168	**0.1161**
3rd scene	1.1571	1.4582	1.3739	**0.1050**
4nd scene	1.3298	1.3845	1.2800	**0.1020**

**Table 2 sensors-21-07630-t002:** Mean Squared error (MSE) of $I_{\phi }$ images.

	Tang’s Method	Holzer’s Method	Nakagawa’s Method	Ours
1st scene	1.8582	1.8288	4.7264	**1.6845**
2nd scene	1.9365	1.9926	4.7174	**1.7900**
3rd scene	2.0040	2.0192	4.8129	**1.8063**
4th scene	2.1487	2.1993	4.8944	**1.9430**

**Table 3 sensors-21-07630-t003:** Structural Similarity Index (SSIM) of $I_{\theta }$ images.

	Tang’s Method	Holzer’s Method	Nakagawa’s Method	Ours
1st scene	0.0008	0.3818	0.4066	**0.6862**
2nd scene	0.0007	0.3828	0.3984	**0.6707**
3rd scene	0.0010	0.3895	0.4092	**0.6776**
4nd scene	0.0011	0.3973	0.4241	**0.6800**

**Table 4 sensors-21-07630-t004:** Structural Similarity Index (SSIM) of $I_{\phi }$ images.

	Tang’s Method	Holzer’s Method	Nakagawa’s Method	Ours
1st scene	0.5278	0.4373	0.1230	**0.6243**
2nd scene	0.5221	0.4354	0.1241	**0.6186**
3rd scene	0.5199	0.4340	0.1247	**0.6154**
4nd scene	0.4942	0.4122	0.1338	**0.5876**

**Table 5 sensors-21-07630-t005:** The full specifications of the implementation environment.

Operating System	MS Windows 10
MATLAB version	2021a
Size of the test image	576 × 640
data type	double precision 64 bit floating point
CPU	Intel CORE i7-3520M @ 2.90 GHz
Memory	8GB DDR3 @ 1600 MHz

**Table 6 sensors-21-07630-t006:** The average execution time for different normal estimation methods for a 576×640 depth image.

Estimation Method	Execution Time (mS)
local plane fitting (considered as ground truth)	5843
Tang’s method [34]	5
Holzer’s method [35]	932
Nakagawa’s method [36]	47
**ours**	**15**

**Table 7 sensors-21-07630-t007:** Extracted radius of different objects (in mm).

	Object 1	Object 2	Object 3	Object 4	Object 5	Object 6
Tran’s Method	28.82	35.44	49.04	33.11	36.42	41.29
Ours	28.53	34.98	53.46	32.88	37.64	43.98
real value	44.88	34.69	46.94	33.22	37.72	54.90

## Data Availability

Not applicable.

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
