# Peer review of "Multiple Cylinder Extraction from Organized Point Clouds"

_sensors, 2021, doi:10.3390/s21227630_

Round 1

Reviewer 1 Report

This is a well-written and structured paper, the investigation results are nicely presented with a proper conclusion.My suggestions are minor, more related to some non-so-relevant details.A literature review might be extended, especially with the potential application of the results, which would be good to discuss too. Figure - maybe add a unit of measure on axis. Figure needs to be positioned according to the text where they are referenced. For example, the figures are in the middle of the conclusion section. Font sizes on images are not consistent due to the printed pages. Line 57 and 64, the sentences starting In [xx], - rewrite.

Author Response

Dear Reviewer,

The comments on your review are addressed in the attached document.

Thank you very much for your time and advice.

Denis Laurendeau

Reviewer 2 Report

This manuscript discusses the extraction of multiple cylinder objects from point clouds. This work demonstrates a thorough investigation and is well-structured in its approach.

However, this manuscript may be improved by stating the following:
1) We see that, based on the results of the experiments, the technique successfully extracts 3-cylinder objects from the input data. However, it is preferable if the research could demonstrate or specifying the number of cylinders that can be detected using this method. Given the title, multiple cylinder extraction, the author can clarify the number of cylinders that can be extracted.
2) Apart from that, how will the spacing/distance between the cylinders effect this technique? This is because the results of the experiments indicate that the cylinders tested are near to the viewing distance. If the author can explain this, it can be used as a reference for other studies.

Author Response

(The authors gave the same response as above.)

Round 2

Reviewer 2 Report

The authors have addressed the points highlighted in the previous review report.